# Electron Beam-Induced Reduction of Cuprite

**Anna Siudzinska** [1,2,3,*], **Sandeep M. Gorantla** [1], **Jaroslaw Serafinczuk** [1,4], **Robert Kudrawiec** [1,5], **Detlef Hommel** [1,2] and **Alicja Bachmatiuk** [1,3,6,*]

1 Łukasiewicz Research Network-PORT Polish Center for Technology Development, Stabłowicka 147, 54-066 Wrocław, Poland
2 Institute of Low Temperature and Structure Research, PAS, 2 Okolna St., 50-422 Wrocław, Poland
3 IFW Dresden, Inst. Complex Mat, 20 Helmholtz Str., D-01069 Dresden, Germany
4 Department of Nanometrology, Faculty of Electronics, Photonics and Microsystems, Wrocław University of Science and Technology, Wybrzeże Wyspiańskiego 27, 50-370 Wrocław, Poland
5 Department of Semiconductor Materials Engineering, Faculty of Fundamental Problems of Technology, Wrocław University of Science and Technology, Wybrzeże Wyspiańskiego 27, 50-370 Wrocław, Poland
6 Key Lab. Adv. Carbon Mat & Wearable Energy Technol, Soochow Inst. Energy & Mat Innovat, Soochow University, Suzhou 215006, China
* Correspondence: anna.siudzinska@port.lukasiewicz.gov.pl (A.S.); alicja.bachmatiuk@port.lukasiewicz.gov.pl (A.B.)

**Abstract:** Cu-based materials are used in various industries, such as electronics, power generation, and catalysis. In particular, monolayered cuprous oxide ($Cu_2O$) has potential applications in solar cells owing to its favorable electronic and magnetic properties. Atomically thin $Cu_2O$ samples derived from bulk cuprite were characterized by high-resolution transmission electron microscopy (HRTEM). Two voltages, 80 kV and 300 kV, were explored for in situ observations of the samples. The optimum electron beam parameters (300 kV, low-current beam) were used to prevent beam damage. The growth of novel crystal structures, identified as Cu, was observed in the samples exposed to isopropanol (IPA) and high temperatures. It is proposed that the exposure of the copper (I) oxide samples to IPA and temperature causes material nucleation, whereas the consequent exposure via e-beams generated from the electron beam promotes the growth of the nanosized Cu crystals.

**Keywords:** electron beam irradiation; cupprite; catalysis; HRTEM

## 1. Introduction

Cu-based materials are of interest in various fields, such as electronics, power generation, and catalysis. Particularly, cuprous oxide ($Cu_2O$) is commonly used as a catalyst in numerous reactions [1,2]. Moreover, $Cu_2O$ is recognized as a p-type semiconductor with a cubic structure (Figure 1), a direct band-gap energy of 2.17 eV, and a high absorption coefficient in the visible region, which makes it an attractive candidate for application in solar cells. Multiple approaches have been used to produce high-efficiency $Cu_2O$-based solar cells [3–5]. Additionally, a one-atom-thick layer of $Cu_2O$ was theoretically investigated and characterized by in situ electron microscopy [6–8]. The synthesis of monolayered $Cu_2O$ with a cubic structure by electron beam irradiation was reported with a theoretical direct band-gap energy of 2.96 eV and rare magnetic properties [7,8], thereby increasing the potential application of $Cu_2O$ in photovoltaics or electronics.

Transmission electron microscopy (TEM) is a powerful tool in characterizing the morphology and properties of various materials, including Cu-based materials. However, copper and its oxides are easily influenced by electron beam irradiation. Their behavior under electron beam radiation has been investigated extensively to elucidate the mechanism of the reversible Cu-$Cu_2O$ redox reaction. Various phenomena were observed, including mass transmission without phase structure variation [9] or the oxidation of nanocrystalline $Cu_2O$ to the CuO phase [10]. Moreover, the transition of copper oxide to copper under

different reducing gas flows, such as hydrogen or methanol, was evaluated by in situ electron microscopy [11–13].

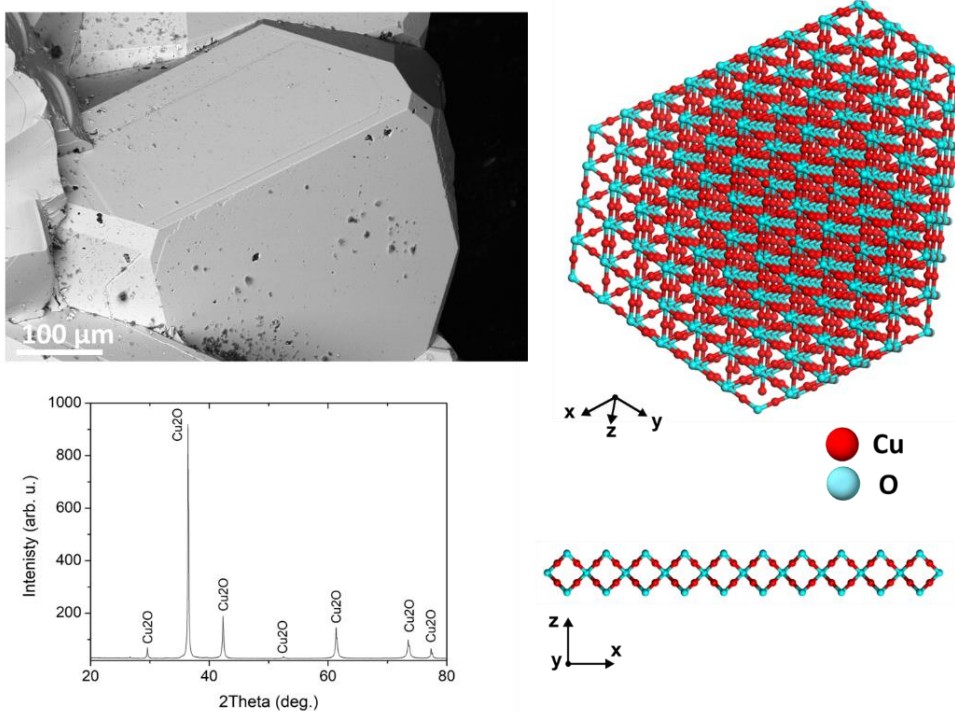

**Figure 1.** SEM micrograph of the bulk $Cu_2O$ crystal (**top left**), XRD pattern of the powdered sample with visible reflections identified as the $Cu_2O$ phase (bottom left), and the crystal structure models of the bulk $Cu_2O$ crystal (**top right**) and theoretical monolayered $Cu_2O$ (**bottom right**).

IPA is known for its reduction ability and is extensively used in the manufacture of Cu-based semiconductors. Thus, knowledge of the mechanism of the reduction process is crucial. The mechanistic and process conditions of the reduction of $Cu_2O$ to copper by isopropanol (IPA) treatment were investigated by Mawaki et al. [14]. Cu surfaces with oxide layers were exposed to IPA gas treatment at different temperatures, and the reduction of copper oxide to copper coupled with IPA decomposition was observed in a temperature-dependent manner [14].

In this study, thin layers of $Cu_2O$ obtained from bulk cuprite crystals are investigated using different methods, including exfoliation in isopropanol, and electron microscopy is used to monitor the effect of electron beam irradiation on the $Cu_2O$ samples. The novelty of our observations is related to more detailed observations of copper (I) oxide reduction in the presence of IPA residues. So far, detailed HRTEM studies using these platforms are not presented within the literature.

## 2. Experimental Methods

### 2.1. Sample Preparation Methods

Thin-layered $Cu_2O$ was obtained from natural bulk cuprite crystals. The bulk sample was analyzed by SEM in its pristine state. A sample of the crystal was ground using a mortar and was characterized by XRD.

The thin-layered $Cu_2O$ was prepared in two ways for TEM analysis. In the first preparation method, the powdered sample was applied onto the lacey carbon TEM grid and subsequently analyzed under the TEM microscope. In the second preparation method, involving the liquid exfoliation approach, a small amount of the powdered sample was suspended in IPA and ultrasonicated for 90 min. Subsequently, the suspension was centrifuged. A small drop of liquid obtained from the top layer of the suspension was applied onto the lacey carbon TEM grid. Before HRTEM analysis, the TEM grids with deposited

material were heated at 120 °C under vacuum conditions ($10^{-5}$ mbar) using a horizontal tube furnace for 14 h to remove residual surface contaminants.

### 2.2. Characterization Methods

#### 2.2.1. SEM and EDS Analysis

The bulk samples were monitored using a Helios NanoLab 450 HP (FEI) Dual-Beam microscope. SEM micrographs were obtained using a secondary and backscattered electron detector at an accelerating voltage of 20 kV and a beam current of 200 pA, with the sample at a working distance of 4 mm. EDS analysis was performed to confirm the chemical composition of the sample. EDS spectra were collected from five different areas of the bulk crystal using a beam energy of 20 keV.

#### 2.2.2. XRD Studies

The powdered sample was measured using an XRD Empyrean diffractometer in the Bragg–Brentano configuration coupled with a PIXcel3D detector in the diffracted beam optical path. XRD reflections were identified using the PDF-4 database, and the identified phase was described by the JCPDS card no. 01-080-7711.

#### 2.2.3. TEM and HRTEM Observation Parameters

TEM and HRTEM analyses were conducted using a double-aberration corrected FEI Titan$^3$ 60–300 (S)TEM microscope equipped with a high brightness X-FEG gun, a Wien-filter monochromator, an image-forming Cs-corrector, a DCOR probe Cs-corrector, ChemiSTEM and super-X EDS detectors, and an EELS spectrometer. Prior to HRTEM analysis, the electron beam conditions were adjusted based on the sample material. In this study, an acceleration voltage of 300 kV was used. The HRTEM measurements were reproduced in a series of six $Cu_2O$ samples, using IPA and heating procedures.

#### 2.2.4. EELS Analysis

EELS analysis was performed in the TEM mode using a Gatan Continuum EELS spectrometer (model 1077) under the following experimental conditions: an operating voltage of 300 kV, an EELS aperture size of 5 mm, and a dispersion of 0.75 eV/ch.

#### 2.2.5. Simulation Techniques

Crystal structure models were generated using Vesta 3 software [15]. Based on the generated models, HRTEM micrograph simulations were conducted using the multi-slice method of the JEMS electron microscopy simulation software. Aberrations of the microscope lenses and current beam corrector settings were considered for the simulations. The intensity profiles of the real and simulated micrographs were evaluated using the ImageJ software.

### 3. Results

### 3.1. SEM, EDS, and XRD of the Bulk Sample

The microstructure of natural bulk $Cu_2O$ (cuprite) was analyzed by SEM to evaluate its quality and purity, and well-defined crystals were observed (Figure 1). X-ray diffraction (XRD) analysis confirmed the chemical and phase compositions of the sample (Figure 1). The XRD reflections shown in Figure 1 were identified as $Cu_2O$ using the PDF-4 database, JCPDS card no. 01-080-7711. Scanning electron microscopy-energy dispersive X-ray spectroscopy (SEM-EDS) elemental composition analysis further confirmed the XRD findings, with strong copper and oxygen signals observed in the EDS spectrum of the sample (supporting information, Figure S1).

We have also performed chemical composition analysis on a local scale using the TEM-EDS method of the samples deposited over TEM grids. The results are presented in Figure S2 (Supporting Information). Namely, four samples over TEM grids were analyzed before and after the heating process, with and without IPA addition. We have only ob-

served signals from Cu, O, and C elements, with different ratios depending on the sample preparation method (Table 1).

**Table 1.** TEM-EDS analysis of $Cu_2O$ samples.

| Element | Dry Transfer, No Heating wt. % | Dry Transfer, Heating wt. % | IPA Transfer, No Heating wt. % | IPA Transfer, Heating wt. % |
|---|---|---|---|---|
| *Copper* | 91.52 | 91.80 | 88.48 | 86.63 |
| *Carbon* | 0.37 | 0.38 | 2.45 | 0.59 |
| *Oxygen* | 8.10 | 7.82 | 9.07 | 9.77 |

### 3.2. HRTEM of Crushed and Exfoliated Sample

High-resolution transmission electron microscopy (HRTEM) measurements were conducted under the optimum conditions to prevent electron beam interference with the analysis. In this study, different electron beam voltages and currents were evaluated. At 80 kV, sample degradation was observed, and imaging was challenging without destroying the sample. In contrast, an electron beam voltage of 300 kV and a low current did not significantly affect the sample. However, the captured micrographs were noisy owing to a low signal (Figure 2). For the $Cu_2O$ sample prepared by crushing the bulk crystal in a mortar, no significant changes were observed during HRTEM analysis at 300 kV. To confirm the stability of the sample under electron irradiation, a series of HRTEM micrographs were analyzed at 300 kV (dose rate = 2970 e$^-$/Å$^2$s) and 80 kV (dose rate = 8330 e$^-$/Å$^2$s) (supporting information, Figure S3).

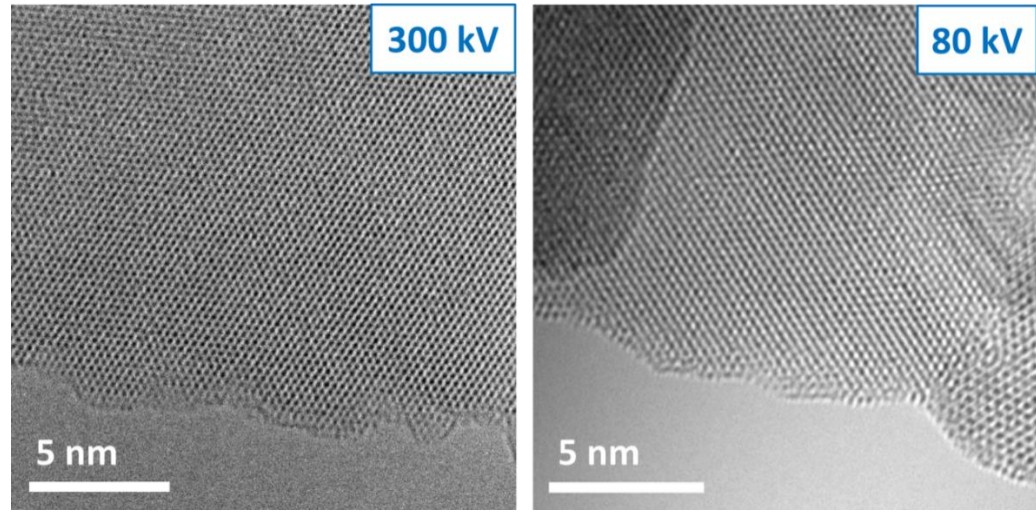

**Figure 2.** HRTEM micrographs of the $Cu_2O$ powdered sample analyzed at different electron beam voltages.

However, when the sample exfoliated in IPA was imaged under the electron beam, rapid structural changes were observed in the material at 80 and 300 kV. To elucidate these phenomena, a series of TEM and HRTEM micrographs were captured at low and high magnifications. The appearance and growth of new structures were observed (Figure 3). The new structure continued to grow from a minimum and maximum Feret particle diameter of 48 and 73.6 nm, respectively, until it stabilized at particle sizes of 70.4 and 104.8 nm, respectively. During the growth stage, the absorption of a nearby particle (indicated by a yellow asterisk in Figure 3) into the growing larger structure was captured by TEM observations (supporting information. Figure S4). Indicative perimeter measurements of both particles were recorded before fusion at 0 s, which were ~118 nm for the small

particle and ~233 nm for the bigger one. After fusion at 75 s, the perimeter of the bigger particle increased to ~383 nm.

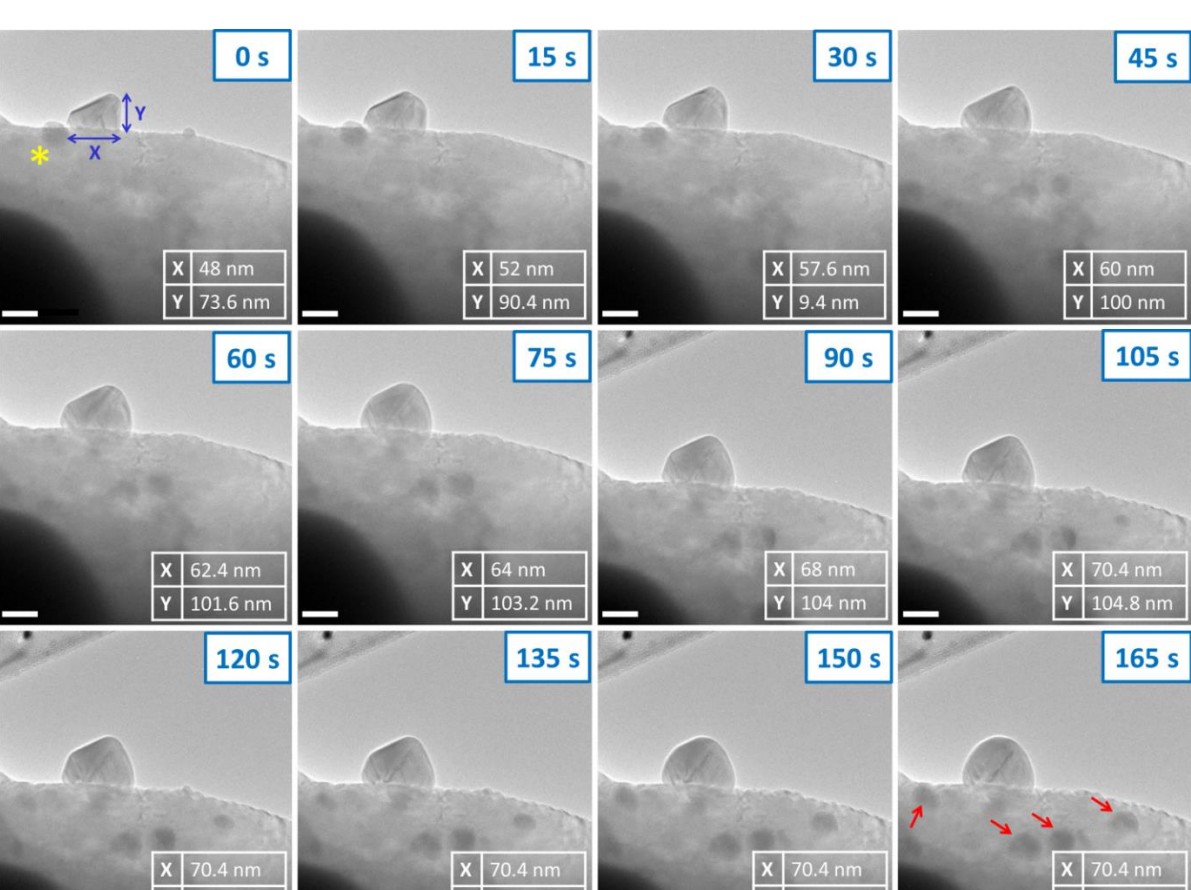

**Figure 3.** TEM image series of the electron beam irradiation effect on the $Cu_2O$ sample exfoliated with isopropanol, with growing structure dimensions measurements. Arrows indicate the appearance of new particles; the asterisk indicates particle absorbed into a larger structure. Scale bars: 50 nm.

During the beam irradiation process, new small particles were observed in the bulk crystal after 45 s, indicated by red arrows in the last frame (Figure 3).

### 3.3. HRTEM Micrograph Simulations of the New Structure

The observed behavior of the $Cu_2O$ sample exfoliated in IPA under electron beam irradiation is presented at a higher resolution using a series of HRTEM micrographs. The time-lapsed micrographs and visualization of the structural growth are shown in Figure 4. An apparent change in the crystal structure and growth of the new crystal was detected in the irradiated material.

Simulation techniques were used to determine the phase of the new structure HRTEM micrographs. The comparison of the intensity profiles of the simulated micrographs of the Cu and $Cu_2O$ crystals with the experimental HRTEM micrograph of the novel particle indicates that the simulated and experimental structures were similar (Figure 5).

### 3.4. EELS Measurements

The chemistry of the sample before and after beam irradiation was evaluated by electron energy loss spectroscopy (EELS) edge analysis. Cu $L_{23}$ near-edge fine structures in the same area before and after irradiation are shown in Figure 6. Sharp Cu peaks, so-called white lines, were only identified in the spectrum recorded before beam exposure. These white lines are characteristic of transition metal oxides, and, in $Cu_2O$, they appear owing to

the hybridization of the Cu and O states [16,17]. Metallic Cu does not exhibit white lines, as observed in the spectrum recorded after beam radiation. Moreover, the O K edge analysis had no oxygen signal in the sample after beam exposure.

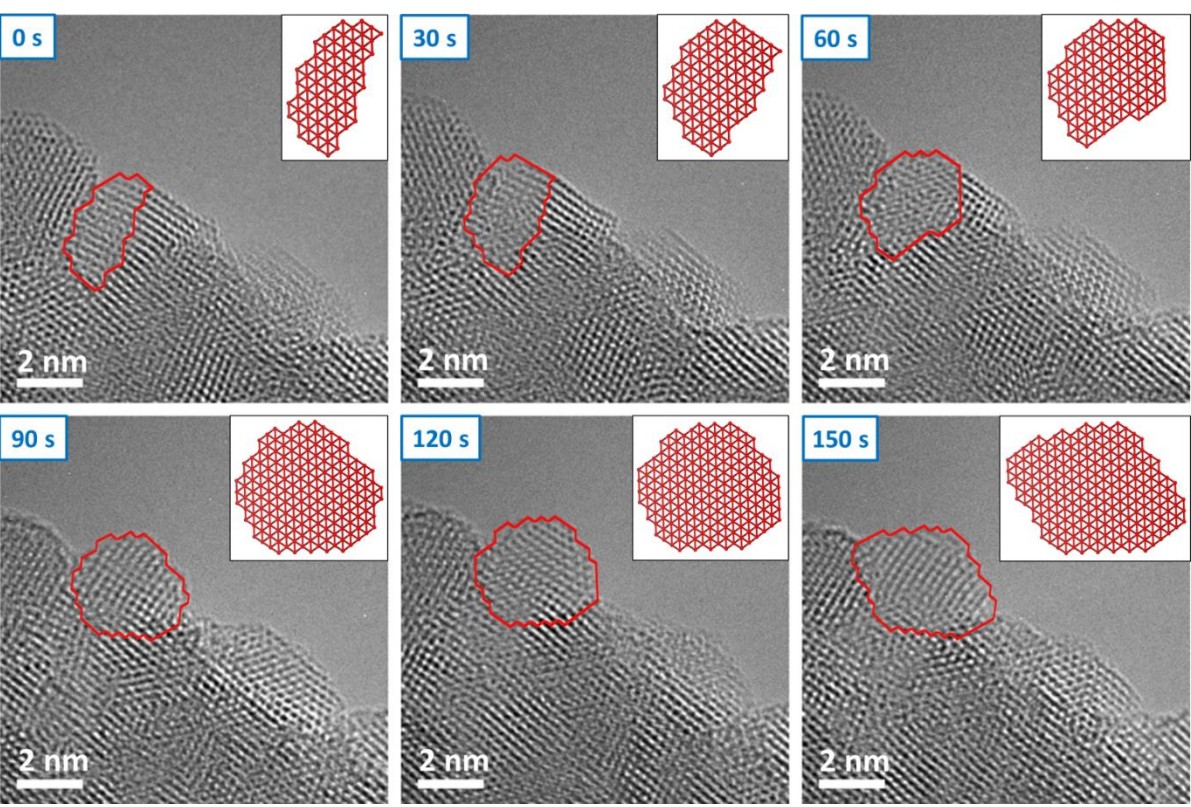

**Figure 4.** HRTEM image series of the beam irradiation effect on the $Cu_2O$ sample exfoliated with isopropanol. The growth of the new structure, identified as Cu, is shown on subsequent images, with an inset of the structure model of the Cu crystal.

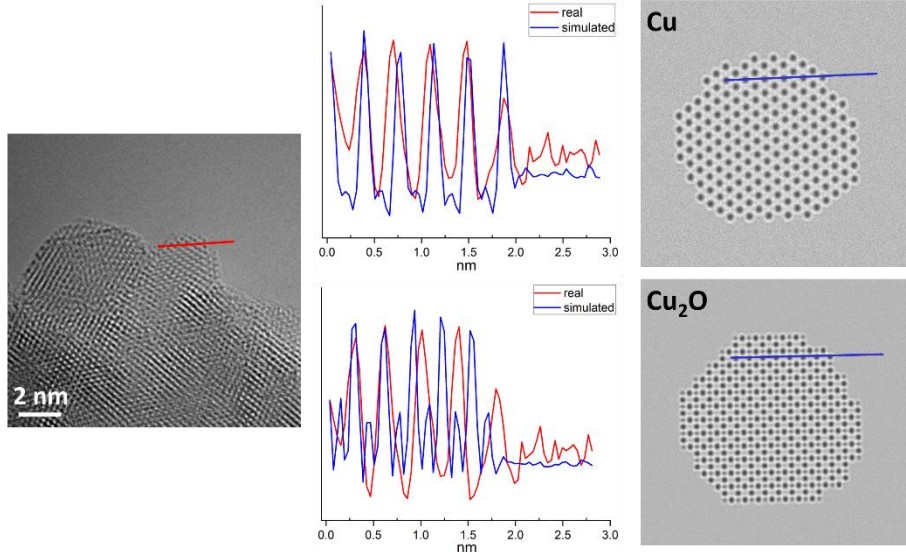

**Figure 5.** Determining the phase of the new structure. Intensity profiles of the simulated HRTEM micrographs of Cu and $Cu_2O$ (**right**) compared with that of the experimental HRTEM micrographs (**left**).

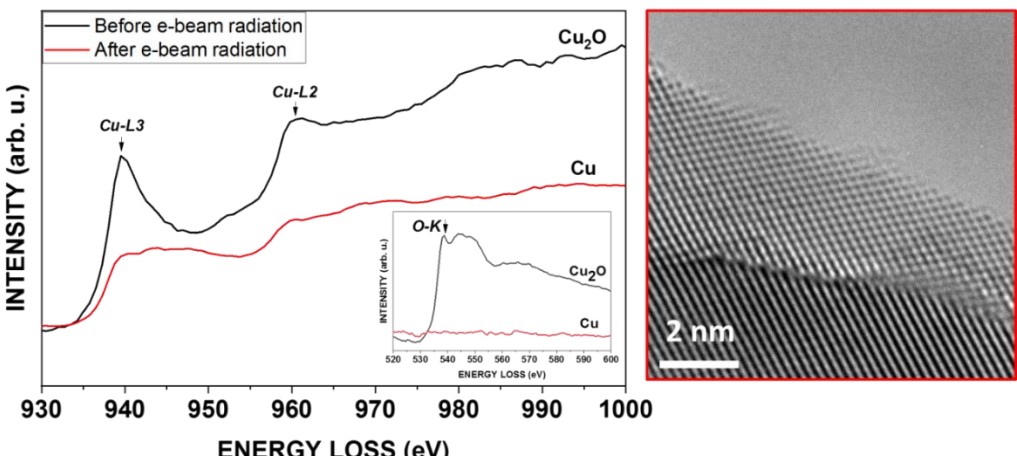

**Figure 6.** EELS spectra of the $Cu_2O$ sample before and after electron beam radiation, Cu spectrum analysis with an inset of the O spectrum, and HRTEM micrograph of the investigated area.

### 3.5. $Cu_2O$ Reduction

Overall, the results of the investigation on the new structure, including simulation techniques such as EELS and fast Fourier transform (FFT) analysis, indicate the reduction of $Cu_2O$ to copper by electron beam radiation. The results confirm the findings reported by Mawaki et al. [14] on the effect of IPA on copper oxide reduction under defined temperature conditions. At temperatures above 100 °C, IPA decomposed and dehydrogenated to form acetone, followed by the generation of $H_2O$ vapor from hydrogen and IPA and oxygen from copper oxide, which resulted in the reduction of copper oxide surfaces [15]. The role of the e-beam irradiation over $Cu_2O$ material is connected to several phenomena (i.e., electron interactions, X-ray irradiation, exposure to specific electron doses at a time) and experimental conditions (e-beam-induced local heating and vacuum conditions).

### 4. Conclusions

This study aimed to perform an HRTEM investigation on the $Cu_2O$ samples obtained from bulk cuprite in the presence of isopropanol residues. In this study, the $Cu_2O$ sample was exposed to IPA during the exfoliation process and subsequently subjected to 120 °C heating in vacuum conditions. The IPA residues on the grid and temperature are assumed to promote the reduction process, which was observed by in situ TEM. The reduction of the copper (I) oxide may be described by the nucleation and growth model [11], whereby small nuclei of the metallic copper phase form within the parent oxide, followed by the growth of the reduced phase. The optimum electron beam parameters for stable starting material observations (300 kV, low-current beam) were selected to prevent beam damage. Samples exposed to IPA and 120 °C formed new crystal structures, identified as metallic copper. We proposed that the exposure of the sample to IPA and heating in vacuum conditions may have caused nucleation, whereas the consequent exposure to the electron beam promoted the growth of the metallic copper crystals.

At the atomic scale, the observed occurrence may be due to the removal of oxygen atoms from their positions by the electron beam and the subsequent structural rearrangement of the lattice. It has been reported that the reduction of copper oxides is based on radiolytic processes of oxygen desorption, and this type of beam degradation is dependent on the beam current density [18]. This agrees well with the observed results because no changes in the samples were observed at low-electron currents.

**Supplementary Materials:** The following supporting information can be downloaded at: https://www.mdpi.com/article/10.3390/met12122151/s1, Figure S1: SEM image and EDS spectrum of the $Cu_2O$ bulk crystal; Figure S2: Electron beam parameters optimization. Effect of different beam voltages on the $Cu_2O$ sample; Figure S3: Beam radiation effect on the $Cu_2O$ sample. Particle

absorption into a larger structure; Figure S4: TEM-EDS spectra and elemental analyses of the $Cu_2O$ samples.

**Author Contributions:** Conceptualization, A.S. and A.B.; methodology, A.S., A.B. and R.K.; software, A.S. and S.M.G.; validation, R.K. and D.H..; formal analysis, A.S., J.S. and S.M.G.; investigation, A.S. and A.B.; resources, A.B.; data curation, A.S.; writing—original draft preparation, A.S. and A.B.; writing—review and editing, A.S. and A.B.; visualization, A.S. and A.B.; supervision, D.H. and A.B.; funding acquisition, A.B., A.S. and J.S. All authors have read and agreed to the published version of the manuscript.

**Funding:** This research was funded by [NCBR] grant number [POIR.04.01.02-00-0103/17-00] and by [NCN] grant number [2021/41/B/ST5/04328] and by and Industrial Doctorate Program from the Ministry of Science and Higher Education in Poland [No. 7/DW/2017/01/1].

**Data Availability Statement:** No application.

**Acknowledgments:** A.S. thanks the National Center for Research and Development (NCBR) project POIR.04.01.02-00-0103/17-00 and the Industrial Doctorate Program from the Ministry of Science and Higher Education in Poland. A.B. and J.S. thank The National Science Center, project 2021/41/B/ST5/04328.

**Conflicts of Interest:** The authors declare no conflict of interest.

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
