# Peer review of "Electron Beam-Induced Reduction of Cuprite"

_metals, doi:10.3390/met12122151_

Round 1
Reviewer 1 Report
The article by Anna Siudzinska et al. "Electron beam reduction of cuprite" presents a study of the formation of new crystal structures in a sample of curpite exposed to gaseous isopropyl alcohol at high temperature. The method of electron beam exposure was used for the study. It is shown that at an electron beam energy of 300 keV, the destruction of the test sample does not occur. The material of the article contains a description of research methods and methods of sample preparation. A brief study is also present. The article may be of interest to the readers of Metals, but it requires significant revision.
1. In the abstract, it should be indicated what is the novelty of the research and what new results are described in the article.
2. What is new in inducing reduction of copper oxide by IPA gas at different temperatures obtained by the authors of the article? What is the role of the electron beam here?
3. In the description of the processing modes, the current value should be indicated, p. 3-4, paragraph 3.2.
4. The conclusion should be reworked. Describe what is new received. On the basis of what results conclusions are drawn.
Reviewer 2 Report
The aim of the present paper is to describe a set of structural methods to characterize the input material in terms of structure. These methods include SEM, EDS, XRD, TEM, HRTEM and others. The results of each method are not presented graphically.
A simulation model of the crystal structure of Cu2O and the possible growth of copper metal is also presented.
Finally, it is stated that the Cu2O sample was annealed. Neither the type of furnace aggregate nor the annealing conditions are given. Whether it took place under normal conditions, under vacuum or reduced pressure.
The aim of this paper is to describe the simulation model of the crystal structure of Cu2O and the growth of a new phase of Cu metal.
The description of the experiment is very general, no schematics of the experimental setup are given. The conditions under which the experiment was carried out are not described. Temperatures, pressure, amount of input material. Voltages other than 80 and 300 kV were used. These values were used based on a literature search or other source of information.
It is reported that the TEM grids were cleaned under vacuum at 120°C overnight. No description is given as to which facility, and no specific time is given. What is overnight?
Chemical analysis before and after the experiment is not given.
Was this a single experiment or was there verification of the result on another series of experiments.
I recommend to continue the research and submit the paper for peer review after the other experimental conditions are completed and completed.
Round 2
Reviewer 1 Report
I am satisfied with the changes made by the authors. The article can be published in this form
Reviewer 2 Report
The authors have made the required modifications and additions. The results of the research are promising and it would be advisable to continue.